**Data Availability Statement:** All relevant data are within the manuscript and its Supporting Information files.

# The increasing incidence of visceral leishmaniasis relapse in South Sudan: A retrospective analysis of field patient data from 2001–2018

Gabriel Naylor-Leyland[1], Simon M. Collin[2,3], Francis Gatluak[4], Margriet den Boer[5], Fabiana Alves[6], Abdul Wasay Mullahzada[1], Koert Ritmeijer[1]*

**1** Médecins Sans Frontières, Amsterdam, The Netherlands, **2** UK Health Security Agency (UKHSA), London, United Kingdom, **3** Departamento de Medicina Social, Universidade Federal do Espírito Santo, Vitória, Brazil, **4** Médecins Sans Frontières, Lankien, South Sudan, **5** Médecins Sans Frontières, London, United Kingdom, **6** Drugs for Neglected Diseases *initiative* (DNDi), Geneva, Switzerland

* koert.ritmeijer@amsterdam.msf.org

## Abstract

### Background

Visceral Leishmaniasis (VL) is endemic in South Sudan, manifesting periodically in major outbreaks. Provision of treatment during endemic periods and as an emergency response is impeded by instability and conflict. Médecins Sans Frontières (MSF) has provided health care in South Sudan since the late 1980's, including treatment for 67,000 VL patients. In recent years, MSF monitoring data have indicated increasing numbers of VL relapse cases. A retrospective analysis of these data was performed in order to provide insight into the possible causes of this increase.

### Methodology/Principal findings

Programme monitoring data from the MSF hospital in Lankien, Jonglei State, South Sudan, for the period 2001–2018 were analysed to detect trends in VL relapse as a proportion of all VL cases presenting to MSF treatment centres. Routinely collected patient-level data from relapse and primary VL cases treated at all MSF sites in South Sudan over the same period were analysed to describe patient characteristics and treatments received. VL relapse as a proportion of all VL cases increased by 6.5% per annum (95% CI 0.3% to 13.0%, p = 0.04), from 5.2% during 2001–2003 to 14.4% during 2016–2018. Primary VL and VL relapse patients had similar age, sex and anthropometric characteristics, the latter indicating high indices of undernutrition which were relatively constant over time. Clinical factors (Hb, spleen size, and VL severity score) also did not vary substantially over time. SSG/PM was the main treatment regimen from 2001–2018, used in 68.7% of primary and 70.9% of relapse VL cases; AmBisome was introduced in 2013, received by 22.5% of primary VL and 32.6% of VL relapse cases from 2013–2018.

**Funding:** The author(s) received no specific funding for this work.

**Competing interests:** The authors have declared that no competing interests exist.

## Conclusion

Increasing incidence of VL relapse in South Sudan does not appear to be explained by changes in patient characteristics or other factors. Our data are concerning and may indicate an emergence of treatment-resistant parasite strains, decreasing the effectiveness of treatment regimens. This warrants further investigation as a causal factor. New chemical entities that will enable safe and highly effective short-course oral treatments for VL are urgently needed.

## Author summary

Visceral leishmaniasis is a fatal parasitic disease which affects impoverished populations living in remote areas. It has caused very high mortality in South Sudan where several major outbreaks have occurred in the past 40 years. In East Africa, the disease can be successfully treated with a combination of injectable sodium stibogloconate and paromomycin given for 17 days. South Sudan was the first country where this regimen was introduced. Now, nearly 20 years later, our data show that it may have become less effective. In our clinic in Lankien, situated in the middle of the endemic area, more and more patients present with relapsed disease and need to be retreated. We have analysed our data and found that the relapse rate as a proportion of all VL cases has increased by 6.5% per annum since 2003. It is possible that this is caused by treatment-resistant parasite strains. Since there is currently no alternative first line treatment, this is of concern. We have planned a prospective observational study where we will follow patients for 12 months in order to determine the current relapse rate, and we highlight the need for new chemical entities that will enable safe and effective short-course oral treatments.

## Introduction

Visceral leishmaniasis (VL, also known as 'kala azar') is a vector-borne disease that is 100% fatal if untreated. Substantial progress has been made in reducing the burden of VL in endemic south Asian countries, with reported cases in India, Bangladesh and Nepal declining from around 80,000 in 1992 to 3,000 in 2019 and elimination becoming a real prospect [1]. By contrast, progress in East Africa has been slow, particularly in countries affected by conflict and political instability, with South Sudan experiencing sustained endemicity and several major VL epidemics over the past 40 years. Médecins Sans Frontières (MSF) is an international medical humanitarian aid organization which has been present in South Sudan before, during and after the civil wars which led to its independence, and has treated 67,000 VL patients since the 1980's. Conflict-related challenges to VL control, including lack of infrastructure and clinical staff, internally-displaced populations, and malnutrition [2], have been compounded recently by the complete withdrawal of UK aid funds for VL control in 2020, the COVID-19 pandemic, and longer-term threats to supplies of diagnostic kits and drugs [3,4]. Pentavalent antimonials have been the mainstay of treatment against VL for many decades, and resistance to these, as already established on the Indian subcontinent by the early 1990s [5], and potentially to newer therapeutic agents such as miltefosine and paromomycin (PM) [6], presents another potential challenge to VL treatment and control.

To avoid monotherapy and shorten treatment duration, first-line 30-day sodium stibogluconate (SSG) monotherapy was phased out in South Sudan from 2003 onwards, replaced by a 17-day injectable combination regimen of SSG and PM [7–9]. Equivalent efficacy of SSG/PM compared to SSG monotherapy was proven in clinical trials in east Africa, with combination therapy showing 91% efficacy [9], and this regimen was adopted in WHO recommendations in 2010 [10] and South Sudan Ministry of Health guidelines in 2011 [11]. Liposomal amphotericin B (AmBisome, Gilead Sciences Inc.) was introduced in the late 90's [12], but its prohibitively high price limited its use to severely ill patients (patients unable to walk or sit up, and with severe weight loss, severe anaemia, severe super-infections, and other markers of poor prognosis). Negotiation of a reduced 'access price' meant that use of AmBisome was gradually expanded from 2012 onwards as second-line treatment and for patients with high intolerance to SSG, including pregnant women, the elderly, and HIV-co-infected patients. AmBisome has been provided as a donation by the manufacturer via WHO for these indications since 2012.

Disease relapse after initial successful treatment of primary VL in a small proportion of patients is a well-described phenomenon, with evidence from trials and field studies indicating a relapse rate of around 5% for SSG/PM combination therapy in East Africa [7,13,14]. Relapse may be attributed to ineffective cellular immunity after treatment related to immunosuppressive conditions such as HIV, TB, and malnutrition, and/or to inadequate treatment resulting in significant persistent parasitaemia despite initial clinical cure. In settings such as South Sudan where active patient follow-up is extremely difficult and not routinely performed, VL relapse rates are monitored passively by VL re-treatment admissions (re-admissions) as a fraction of the total number of VL admissions. In the past few years, this passive monitoring has indicated an apparent increase in re-treatment rates. The aims of the present study were to analyse programme data to determine trends in VL relapse rates and to use patient-level data to generate hypotheses which might explain any such trends.

## Methods

### Ethics statement

This research fulfilled the exemption criteria set by the MSF Ethical Review Board (ERB) (https://scienceportal.msf.org/assets/7002) for retrospective analysis of routinely-collected clinical data from pre-existing, established programs, and thus did not require MSF ERB review (https://scienceportal.msf.org/assets/6996). It was conducted with permission from the Medical Director of the MSF Operational Centre Amsterdam.

### Study design, setting and population

Due to the dynamic epidemiology of VL in South Sudan, characterised by large fluctuations in case numbers between locations and from year to year, with focal and seasonal outbreaks, MSF has a flexible operational model that has included provision of treatment at more than 25 sites across South Sudan, in Unity, Jonglei, and Upper Nile states. VL patient numbers in MSF's facilities over the past 10 years haved averaged over a thousand patients per year, ranging from a few hundred to over 6000 patients yearly depending on the epidemiology. This retrospective analysis used routine clinical data collected mainly from a Médecins Sans Frontières (MSF) treatment facility in Lankien, Jonglei State, in the north eastern part of South Sudan. This facility has been run by MSF since 1993, providing primary outpatient and inpatient health care services, comprehensive maternity care, and treatment for TB, HIV, and tropical diseases including VL. The catchment area population is mainly pastoralist, practicing cattle husbandry and agriculture. The facility is located in an area with high VL endemicity and 50.2% of all VL patients treated by MSF during the study period were treated here, and

75% of VL patients treated during 2013–2018. Between 2013–2018, 66% of all reported VL cases in South Sudan were treated by MSF (50% in Lankien).

Access to the facility is often difficult because of poor road conditions, flooding during the rainy season, and insecurity.

### Diagnosis and treatment of VL in South Sudan

Clinical management followed WHO and South Sudan Ministry of Health guidelines [10,11,15] as summarised below.

*Diagnosis*. All patients presenting with a history of fever of more than 2 weeks and splenomegaly and/or lymphadenopathy and/or wasting were considered VL suspects and underwent further diagnostic evaluations. Patients without prior VL treatment history (suspect primary VL) were screened using the rK39 rapid diagnostic test (IT-Leish, Bio-Rad laboratories, USA) with a positive result considered confirmatory. Those testing negative were screened with the leishmania direct agglutination test (DAT, Royal Tropical Institute, Amsterdam, The Netherlands), with a high titer (≥1:6400) considered confirmatory. Those with an intermediate DAT titer (1:800–1:3200) underwent tissue aspiration (spleen or lymph node), with a positive result confirming VL. Patients with prior VL treatment history (suspect relapse VL) underwent tissue aspiration (mostly splenic) with diagnosis by microscopy. Routine testing of all primary and relapse VL patients for HIV commenced in 2012.

*Treatment*. First line treatment was SSG (20 mg/kg) in combination with PM (15 mg/kg) given on an ambulatory basis over 17 days with daily intramuscular injections. Patients contra-indicated for SSG (pregnant women), with known poor tolerability of SSG (age >45 years, severe VL), and patients who did not respond to SSG were treated with AmBisome by 6 IV infusions of 5 mg/kg on alternate days. Relapse patients were treated either with SSG (20 mg/kg for 30 days) in combination with PM (15 mg/kg for 17 days) or AmBisome (6 IV infusions of 5 mg/kg on alternate days) [16,17]. Patients with HIV/VL co-infection received AmBisome. Patients with TB/VL typically have other symptoms qualifying as severe VL and are therefore often treated with AmBisome. The drugs were from the same quality-controlled source during the entire period (paromomycin: Gland Pharma, India; SSG: Albert David, India; AmBisome: Gilead, USA).

*Discharge*. Patients with an increased risk of treatment failure or relapse, i.e. patients with a prior episode of VL, or patients with inadequate or doubtful clinical response, required a negative parasitological test-of-cure by aspirate microscopy to confirm cure (at 21 days for primary VL, at 30 days for VL relapse). All other patients were discharged without test-of-cure. Patients were not actively followed up but were advised to come back to the clinic in case symptoms re-occurred.

*Treatment adherence and default*. Treatment adherence was encouraged through incentives, with patients receiving food rations, PlumpyNut supplement and mosquito nets when they completed their course of injections. Doses of SSG, PM or AmBisome were checked independently by the programme supervisor after weighing the patient on admission and every week thereafter, and the number of injections and doses were recorded in treatment records. A defaulter protocol applied throughout the study period whereby patients who missed two consecutive visits were actively traced to ensure that they completed their course of treatment.

### Patient-level data

Patient-level data were obtained from a pseudonymised archive of routinely-collected clinical data from all MSF treatment centres in South Sudan from 2001–2018. Demographic, diagnostic, treatment, and discharge information were handwritten on cards from which data were

double checked before subsequent entry into electronic databases, which were merged for the purpose of this analysis. Patient cards from various treatment sites were lost during periods of acute armed conflict when MSF staff had to be evacuated and health facilities were looted or burned and, on other occasions, stored patient cards were lost during flooding. For these reasons, some patient-level data were missing, especially from peripheral sites. Potentially duplicate records created by merging databases were identified by combining patient number, treatment site, sex, and date of admission.

A relapse case was defined as any patient who was recorded with a diagnosis of 'Relapse VL' on admission, based on the attending medical staff asking the patient whether they had previously been treated for VL more than 30 days before admission for the current VL episode.

We used all available patient-level data to analyse primary and relapse VL patient characteristics (rather than restricting only to patients treated in Lankien) to maximise the sample size and because treatment site wasn't routinely recorded in the first year of the study period. To simplify comparisons of patient characteristics over time, we aggregated patient-level data into 3-year periods (2001–2003, 2013–2015, and 2016–2018) except for the years 2004–2012, which saw relatively low numbers of VL patients. These periods also broadly correspond with the introduction of SSG/PM combination therapy (from 2003/2004) and AmBisome (from 2012).

Patient-level clinical data included weight-for-height Z-score for children aged ≤10 years, BMI-for-age Z-score for 11–19 year-olds, BMI for adults aged 20+ years, haemoglobin (Hb), spleen size (distance in centimetres from left costal margin to tip of spleen). A VL severity score was derived from risk factors for VL mortality: age, nutritional status (weight-for-height Z-score for children and adolescents aged <19 years, BMI for adults aged 19+ years), haemoglobin (Hb), and level of weakness. Patients with a risk score associated with >20% probability of death were classified as having 'severe' VL and received AmBisome, aggressive antibiotic treatment and nutritional care. Time-to-relapse was calculated for VL relapse patients whose discharge date for their primary VL episode was recorded in the database.

## Monitoring data

Programme monitoring data include aggregate counts of cases by treatment site, diagnosis (primary VL, VL relapse, and post-kala azar dermal leishmaniasis (PKDL)) and outcome. For the purpose of analysing trends in relapse, we used programme monitoring data only from Lankien because it is the only treatment site that has provided VL services throughout the study period (except from 06/2004-06/2005 when staff were evacuated for security reasons) and it has the most consistent and continuous recording of VL diagnostic data (S1 Table).

## Statistical methods

Trends in VL relapse as a proportion of all VL cases (primary and relapse) recorded in programme monitoring data from 2001–2018 were analysed using Joinpoint trend analysis software (Joinpoint Regression Program, Version 4.7.0.0—February 2019; Statistical Methodology and Applications Branch, Surveillance Research Program, National Cancer Institute, Bethesda MD, USA). Trends were estimated as the annual percentage change (APC) by fitting models linear on the log of the percentage VL relapse with minimum 0 joinpoints, maximum 3 joinpoints, minimum 2 observations between joinpoints, 4,500 Monte Carlo permutations, and $\alpha$ = 0.05 [18]. Differences in characteristics of primary and relapse VL cases from patient-level data over time were tested using Chi-squared tests for categorical variables and non-parametric K-sample test on the equality of medians, non-parametric test-for-trend, and Student's t test (if normally distributed) for continuous variables, all with $\alpha$ = 0.05. Logistic regression was used to estimate odds ratios for VL relapse by HIV and TB status, adjusted for age and sex.

## Results

### Trends in VL relapse from MSF programme monitoring data

The proportions of all VL patients in Lankien who were diagnosed as primary VL or relapse VL for each year from 2001 to 2018 are summarised in **Table 1**. Join-point analyses indicated that the best-fit model had zero join points and an annual percent change in relapse as a proportion of all VL cases from 2001–2018 of 6.5% (95% CI 0.3% to 13.0%, p = 0.04) (**Fig 1**).

### Primary and relapse VL patient characteristics

The flow of routinely-collected patient-level data from clinical records is shown in **Fig 2**. VL relapse as a proportion of all VL cases in patient-level data was consistent with programme monitoring data from Lankien for most of the study period, with VL relapse patients representing 6.2% of all VL patients in 2001–2003, 11.2% in 2013–2015 and 15.0% in 2016–2018. We compared this with programme data (**Table 1**) where we found 5.2%, 11.4% and 14.4% for the same time periods. During 2004–2012, the proportion of relapses in patient-level data from all sites (7.3%, 145/2,001) was lower than in programme data from Lankien (9.4%, 360/3,381), suggesting that relapse cases were more reliably diagnosed or recorded in Lankien and/or that previously treated patients were more likely to return to Lankien for re-treatment and/or that records missing from this period were more likely to be for readmitted patients.

VL patients (primary and relapse) were mainly children and adolescents: 26% of all patients were ≤5 years old; 66% were ≤16 years old. The median (IQR) age was 12 (5–22) years. There was a slight predominance of males (51.6%). These demographic characteristics were fairly constant over time (**Table 2**), albeit with an increase in the proportion of patients age ≤16 years from 61.1% in 2001–2003 to 68.9% in 2016–2019 and a peak in the male-to-female ratio (55.0% male) in 2013–2015.

Anthropometric indices and haemoglobin levels were consistent with high prevalence of undernutrition and anaemia in the population throughout the study period (**Table 2**), with a

**Table 1. Primary and relapse visceral leishmaniasis (VL) cases as proportions of all VL patients treated at Médecins Sans Frontières (MSF) centre in Lankien, South Sudan, 2001–2018 (from programme monitoring data).**

| Year | Total patients treated | Primary VL (%) | VL relapse (%) |
|---|---|---|---|
| 2001 | 634 | 615 (97.0%) | 19 (3.0%) |
| 2002 | 1451 | 1420 (97.9%) | 31 (2.1%) |
| 2003 | 2391 | 2207 (92.3%) | 184 (7.7%) |
| 2004 | 504 | 422 (83.7%) | 82 (16.3%) |
| 2005 | 784 | 746 (95.2%) | 38 (4.9%) |
| 2006 | 438 | 356 (81.3%) | 82 (18.7%) |
| 2007 | 144 | 127 (88.2%) | 17 (11.8%) |
| 2008 | 61 | 53 (86.9%) | 8 (13.1%) |
| 2009 | 179 | 172 (96.1%) | 7 (3.9%) |
| 2010 | 1010 | 964 (95.5%) | 46 (4.6%) |
| 2011 | 493 | 437 (88.6%) | 56 (11.4%) |
| 2012 | 218 | 194 (89.0%) | 24 (11.0%) |
| 2013 | 1338 | 1288 (96.3%) | 50 (3.7%) |
| 2014 | 4589 | 4202 (91.6%) | 387 (8.4%) |
| 2015 | 1399 | 1004 (71.8%) | 395 (28.2%) |
| 2016 | 1425 | 1299 (91.2%) | 126 (8.8%) |
| 2017 | 1786 | 1567 (87.7%) | 219 (12.3%) |
| 2018 | 1003 | 742 (74.0%) | 261 (26.0%) |

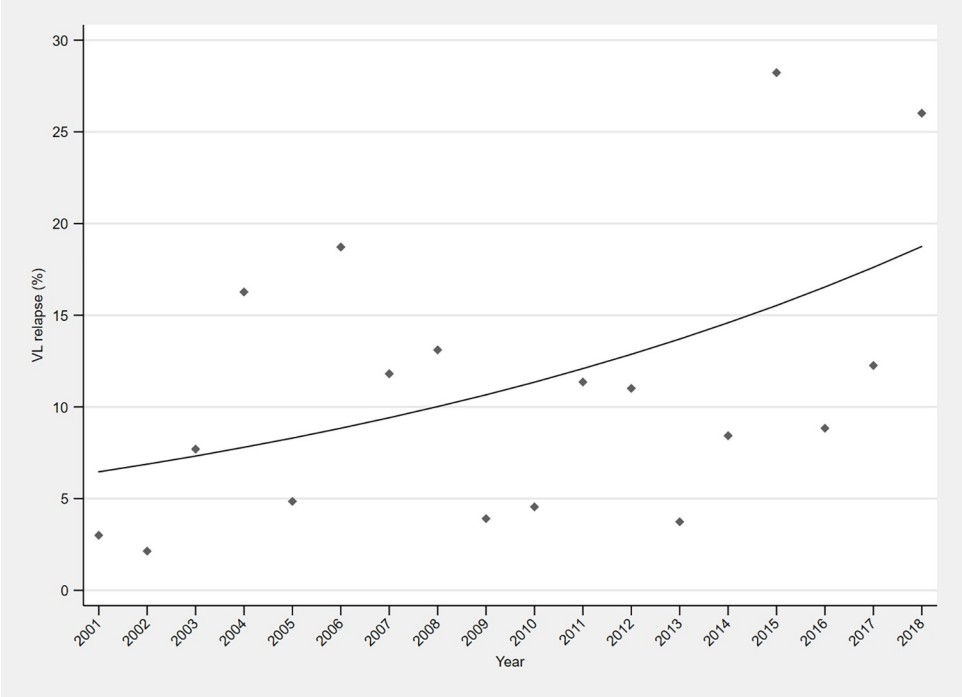

**Fig 1. Trend in relapse as a proportion of visceral leishmaniasis (VL) cases treated at Médecins Sans Frontières (MSF) centre in Lankien, South Sudan, 2001–2018 (from programme monitoring data)**\*. \* Join-point analyses indicated that the best-fit (log linear) model had zero join points and an annual percent change in relapse as a proportion of all VL cases from 2001 to 2018 of 6.5% (95% CI 0.3% to 13.0%, p = 0.04).

mean (SD) weight-for-height Z-score of -1.6 (1.5) in the ≤10-year-old age group, mean BMI-for-age Z-score of -3.0 (1.1) in 11–19 year-olds, mean BMI of 15.7 (2.1) kg/m$^2$ in adults aged 20 years and older, and mean Hb of 9.2 (2.0) g/dL. Variation among primary VL patients suggested that anaemia and nutritional status of children ≤10 years old improved from 2001–2003 to 2013–2015 before worsening again in 2016–2018 and the nutritional status of older children and adults improved slightly overall. VL illness severity score was slightly higher in primary VL patients in 2001–2003 otherwise constant over time in both patient groups.

VL relapse patients did not differ from primary VL patients by median age (p = 0.76) or mean BMI for adults age 20+ years (Student's t test p = 0.77). There was a slightly higher proportion of males (53.9% vs. 51.4%, p = 0.03) and VL relapse patients age ≤10y had slightly worse mean weight-for-height Z-score (-1.67 vs. -1.57, Student's t test p = 0.05) and slightly better mean BMI-for-age Z-score (-2.88 vs. -3.01, Student's t test p = 0.03).

## Treatments and outcomes for primary VL and VL relapse patients

SSG/PM was the main treatment regimen from 2001–2018, used in 72.2% (13,772/19,063) of primary VL and 73.2% (1,509/2,061) of VL relapse cases (**Table 3**). SSG monotherapy was largely phased out by 2003. After 2012 AmBisome was used to treat 22.5% (2,245/9,986) of primary VL and 32.6% (470/1,440) of VL relapse cases from 2013–2018. Treatment default rates were consistently low over the years: 1.3% in 2001–2003; 1.6% in 2004–2012; 1.5% in 2015–2015; and 0.3% in 2016–2018.

Time from end of primary VL treatment to re-admission for VL relapse was constant over the study period (p = 0.12), with 66.9% (975/1,457) of all relapses for which time-to-relapse

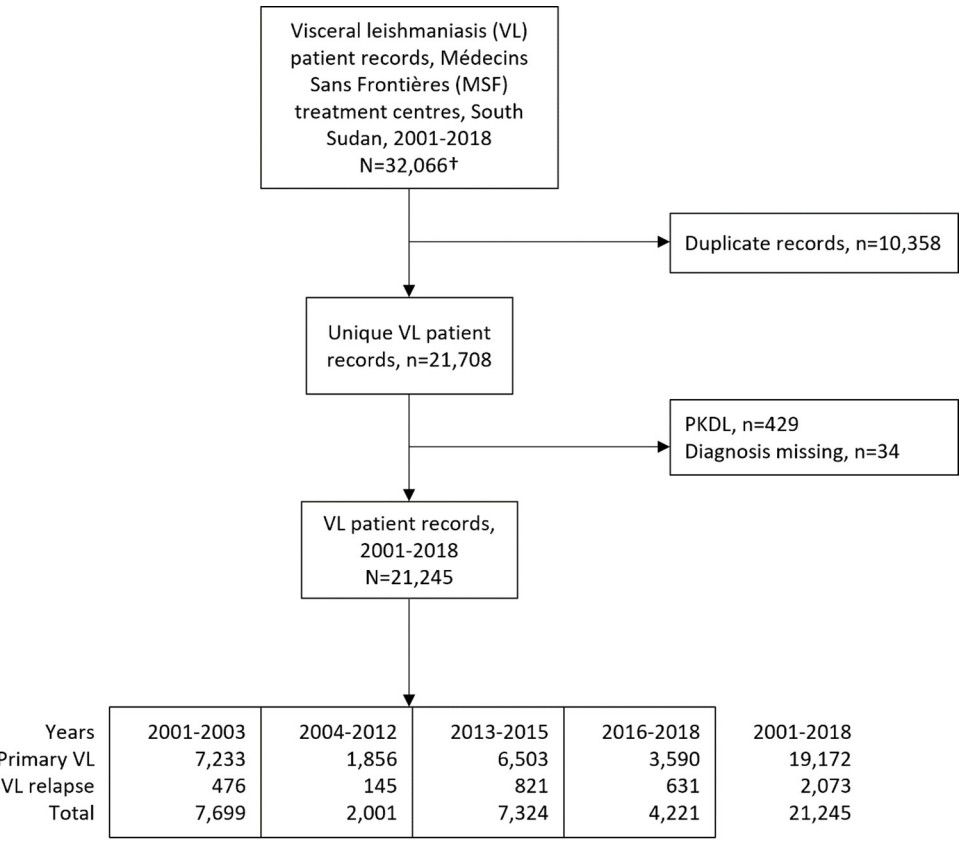

† Routinely-collected data were collated from several sources into a single spreadsheet hence, the need for de-duplication. This was done by dropping duplicates of a patient identifier comprising patient number + treatment site code + sex + date of admission

**Fig 2. Flow of routinely-collected patient-level data from clinical records.**

data were available occurring within 6 months. Nevertheless, 15.3% (223/1,457) of relapses occurred more than one year after primary VL treatment (**Table 3**).

As with anthropometric indices and haemoglobin levels, spleen sizes on admission improved slightly over the study period and the proportion of VL patients discharged without detectable splenomegaly increased from 88.6% (3,998/4,511) in 2014 to 99.2% (904/911) in 2018 (**Table 4**).

A small number (median 5, range 1–7) of HIV co-infected patients were detected each year from 2012 onwards, equivalent to an annual HIV co-infection rate in the years 2013–2018 of 0.1–0.4% (3.7% (7/189) in 2012) with no discernible trend. The proportion of relapse VL patients with a concomitant HIV infection (1.03%, 15/1,457) was more than six times higher than the HIV co-infection rate in primary VL patients (0.14%, 14/10,156), corresponding to an age-sex adjusted odds ratio of 6.58 (95% CI 3.15, 13.8) or 6.74 (3.22, 14.1) if patients with no HIV test result were considered to be negative. Similarly, the proportion of relapse VL patients with a confirmed concomitant tuberculosis (TB) infection (0.96%, 14/1,459) was more than four times higher than in primary VL patients (0.24%, 24/10,144), age-sex adjusted odds ratio 4.02 (95% CI 2.07, 7.81) or 4.09 (2.11, 7.93) if patients with no HIV test result were considered to be negative.

**Table 2. Characteristics of primary visceral leishmaniasis (VL) and VL relapse patients treated at Médecins Sans Frontières (MSF) centres in South Sudan, 2001–2018 (patient-level data on admission).**

| | 2001–2003 | 2004–2012* | 2013–2015 | 2016–2018 | p-value† |
|---|---|---|---|---|---|
| Primary VL | n = 7,223 | n = 1,856 | n = 6,503 | n = 3,590 | |
| Age (years), median (IQR) | 12 (6–25) | 12 (5–25) | 10 (5–20) | 13 (6–19) | <0.001 |
| Sex (male), % | 48.5% | 54.3% | 54.5% | 50.1% | <0.001 |
| Weight-for-height WHO Z-score, age ≤10y, mean (SD) | -1.93 (1.36), n = 3,027 | -1.58 (1.40), n = 792 | -1.23 (1.51), n = 3,187 | -1.55 (1.49), n = 1,219 | <0.001 |
| BMI-for-age WHO Z-score, age 11-19y, mean (SD) | -3.14 (1.08), n = 1,123 | -3.01 (1.02), n = 299 | -2.93 (1.13), n = 1,375 | -2.95 (1.11), n = 1,303 | 0.001 |
| BMI, age >19y, mean (SD) | 15.5 (2.0), n = 2,224 | 15.5 (2.0), n = 545 | 16.0 (2.1), n = 1,635 | 16.0 (2.1), n = 858 | <0.001 |
| Hb (g/dL), mean (SD) | 7.8 (2.2), n = 1,095 | 8.9 (2.4), n = 802 | 9.5 (2.0), n = 6,462 | 9.2 (2.0), n = 3,527 | <0.001 |
| VL severity score (range 0–14), median (IQR) | 3 (2–4), n = 1,048 | 2 (1–3), n = 1,029 | 2 (1–3), n = 6,503 | 2 (1–3), n = 3,591 | <0.001 |
| VL relapse | n = 476 | n = 145 | n = 821 | n = 631 | |
| Relapse as % of all VL patients | 6.2% | 7.3% | 11.2% | 15.0% | <0.001 |
| Age (years) | 12 (6–25) | 13 (5–27) | 10 (5–23) | 13 (5–24) | 0.01 |
| Sex (male) | 46.7% | 50.0% | 59.6% | 53.0% | <0.001 |
| Weight-for-height WHO Z-score for age ≤10y, mean (SD) | -1.94 (1.39), n = 195 | -1.78 (1.32), n = 61 | -1.52 (1.52), n = 412 | -1.69 (1.73), n = 233 | 0.09 |
| BMI-for-age WHO Z-score for age 11-19y, mean (SD) | -2.89 (1.12), n = 90 | -2.94 (1.15), n = 18 | -2.96 (1.22), n = 133 | -2.87 (1.17), n = 160 | 0.93 |
| BMI, age >19y, mean (SD) | 15.9 (2.1), n = 135 | 15.7 (2.1), n = 56 | 15.7 (2.2), n = 230 | 15.8 (1.9), n = 171 | 0.61 |
| Hb (g/dL), mean (SD) | 9.1 (2.3), n = 195 | 8.6 (2.4), n = 99 | 9.0 (2.0), n = 819 | 8.8 (1.9), n = 593 | 0.12 |
| VL severity score (range 0–14), median (IQR) | 2 (1–4), n = 163 | 2 (1–4), n = 101 | 2 (1–4), n = 821 | 2 (1–3), n = 630 | 0.14 |

* Data for 2004–2012 were combined because there were relatively low numbers of patients treated each year during this period, recording of patient-level data was not continuous and some records were lost as a result of attacks on MSF treatment centres

† Non-parametric K-sample test on the equality of medians for continuous variables; Chi-squared test for categorical variables

## Discussion

### Key finding and interpretation

Our analysis has shown that rates of VL relapse in South Sudan increased at a steady rate between 2001 and 2018, with a 2.4-fold overall increase in relapse as a percentage of all VL cases from 6% in 2001–2003 to 15% in 2016–2018.

An earlier analysis of MSF patient-level data (from 1999–2007) showed that age, sex, and malnutrition were not associated with risk of VL relapse, but a positive association with increased spleen size on admission and discharge was reported [19], whilst a slower rate of

**Table 3. Treatment of visceral leishmaniasis (VL) patients and time to relapse at Médecins Sans Frontières (MSF) centres in South Sudan, 2001–2018 (from patient-level data).**

| | | 2001–2003 | 2004–2012* | 2013–2015 | 2016–2018 |
|---|---|---|---|---|---|
| Primary VL treatment | | n = 7,223 | n = 1,856 | n = 6,503 | n = 3,590 |
| | SSG | 2,683 (37.2%) | 140 (7.5%) | 2 (0.0%) | 0 (0.0%) |
| | SSG/PM | 4,388 (60.8%) | 1,655 (89.2%) | 5,042 (78.1%) | 2,687 (76.1%) |
| | AmBisome | 130 (1.8%) | 59 (3.2%) | 1,401 (21.7%) | 844 (23.9%) |
| | Other | 20 (0.3%) | 2 (0.1%) | 9 (0.1%) | 1 (0.0%) |
| Time to VL relapse (days) | median (IQR), n | 139 (98–220), n = 248 | 118 (82–178), n = 80 | 126 (85–228), n = 777 | 132 (82–349), n = 352 |
| | 21–90 days | 49 (19.8%) | 25 (31.3%) | 221 (28.4%) | 105 (29.8%) |
| | 91–180 days | 115 (46.4%) | 36 (45.0%) | 312 (40.2%) | 112 (31.8%) |

* Data for 2004–2012 were combined because there were relatively low numbers of patients treated each year during this period, recording of patient-level data was not continuous and some records were lost as a result of attacks on MSF treatment centres

**Table 4. Spleen size (distance in centimetres from left costal margin to tip of spleen) on admission and discharge for primary and relapse visceral leishmaniais (VL) patients at Médecins Sans Frontières (MSF) centres in South Sudan, 2014–2018 (from patient-level data).**

| Primary VL | 2014 | 2015 | 2016 | 2017 | 2018 | p-value* |
|---|---|---|---|---|---|---|
| Admission (median (IQR)) | 2 (0–5), n = 4,119 | 4 (0–6), n = 978 | 2 (0–5), n = 1,284 | 2 (0–4), n = 1,562 | 2 (0–4), n = 686 | 0.008 |
| Spleen 0 on discharge | 89.0% (3,665/4,117) | 92.9% (911/981) | 92.1% (1,181/1,283) | 93.8% (1,464/1,561) | 99.4% (679/683) | <0.001 |
| VL relapse | | | | | | |
| Admission (mean (SD)) | 4 (0–8), n = 397 | 4 (2–8), n = 410 | 4 (2–6), n = 142 | 4 (1–6), n = 228 | 4 (0–6), n = 228 | 0.05 |
| Spleen 0 on discharge | 84.5% (333/394) | 94.4% (387/410) | 83.1% (118/142) | 94.7% (216/228) | 98.7% (225/228) | <0.001 |

* Non-parametric test-for-trend across years for spleen size as continuous variable; Chi-squared test for spleen size 0 on discharge

regression of splenomegaly was positively associated with risk of VL relapse in India [20]. In our data, spleen size was recorded consistently only from 2013 onwards, therefore, we could not investigate trends over the whole study period. However, between 2014–2018 there was no substantial variation in median spleen sizes on admission, and the proportion of primary VL patients discharged with a normal spleen increased from 89% to 99%.

Low Hb levels were reported to be associated with increased risk of VL relapse in Brazil [21] and Georgia [22] but not in India [20]. We found that median HB levels on admission increased between 2001–2018, reflecting a general improvement in the nutritional status of older children and adults albeit from a very low baseline. The nutritional status of children ≤10 years old improved from 2001–2003 to 2013–2015 but appeared to worsen again in 2016–2018, which gives cause for concern in terms of risks to child survival and to child growth and development. Co-infection with HIV and TB are strong risk factors for relapse, increasing the odds of relapse 7-fold and 4-fold, respectively, as seen in our data and elsewhere [23], but prevalence of these diseases has remained low (< 1%) in this population, with no observed trend. A VL severity score used operationally by MSF was slightly higher in primary VL patients in 2001–2003 but was otherwise static. Compliance to treatment was high throughout.

These relatively constant patient characteristics suggest that the substantial increase in incidence of VL relapse in this population must be explained by other factors. In the absence of evidence for changes in the susceptibility of the population or in the pathogenicity of *L. donovani* in East Africa, we must consider reduced effectiveness of treatment related to drug sensitivity. We must also exclude other factors as an explanation, including improvements in admission procedures or data recording leading to more re-admitted patients being identified correctly as VL relapse cases and improvements in security and health provision enabling more VL relapse cases to return for treatment.

We would expect better access to VL treatment to apply to all cases, primary and relapse therefore, this would be unlikely to bias our calculation of the latter as a proportion of all cases. Similarly, we cannot attribute the increase in VL relapse to larger numbers of older patients who had primary VL further back in time, because patient age (71% under 20 years old) and time-to-relapse (67% within 6 months) were constant in our data. Neither did our data indicate higher apparent VL relapse rates related to easier access to treatment centres during the relatively peaceful years between the 2005 peace treaty and the outbreak of civil war in 2013, although total case numbers were lower during this period than in other years. A higher proportion of relapses can also be expected to be observed in the year following a relatively higher number of VL cases because, in the second year, the number of new VL cases decreases and the number of relapses proportionally increases. We can discern this in our programme data for 2004 (16% VL relapse) following the 2002/2003 outbreak, 2011/2012 (11%) following the 2010 outbreak, and 2015 (28%) following the 2014 outbreak. The steady upward trend in VL

relapse cases superimposed on these and other short-term fluctuations in our data was supported by moderate statistical evidence, and future analysis using a longer time series might discern correlation with background VL incidence, seasonality, and other ecological factors.

## Implications for policy and practice

If resistance has emerged for paromomycin, a drug for which resistance is easily induced *in vitro* [24], and/or for SSG, for which widespread resistance was demonstrated in India after decades of use, then treatment with 17 days SSG/PM will leave many patients parasitaemic despite a good initial clinical response, who will then be at increased risk of relapse. This will further increase the probability of development of resistant strains for either component of the combined regimen. There is also some evidence that the current regimen of 30mg/kg AmBisome is not fully effective in East African VL [25], with many patients requiring extended treatment. Although this regimen is recommended by WHO as a second-line treatment of VL caused by *L. donovani* in East Africa, evidence for this regimen is lacking. The regimen of 30mg/kg was decided somewhat arbitrarily, based on studies for treatment of VL caused by *L. infantum* in the Mediterranean [26,27]. Later studies with *L. donovani* in East Africa showed high failure rates in dose-finding studies which only went up to 21mg/kg [28].

High retreatment rates have not been reported from other East African countries [13,14]. However, the SSG/PM combination regimen has only been rolled out in other countries more recently (after 2010), whereas it was introduced in the MSF programmes in South Sudan in 2002–2003. *L. donovani* parasites in our study area have therefore been exposed to the combination regimen for at least 7 years longer than in other countries, indicating that our data from South Sudan might provide early warning for the emergence of parasite strains with decreased drug sensitivity.

## Limitations

The main limitation of our study is that routinely-collected demographic and clinical data from resource-poor and conflict-affected settings will be subject to errors of measurement, recording and classification, although these are more likely to be random than systematic and will therefore tend to bias estimates towards the null. Treatments were standardised (SSG17/PM17 for primary VL, SSG30/PM17 for VL relapse, 6 x 5mg/kg AmBisome for both) and protocols implemented to ensure adherence and minimise default, but patient-level recording of the precise treatment received (dose and duration), test-of-cure and whether patients were re-admitted for VL relapse are subject to error. VL relapse patients were usually issued with new patient numbers because of the very similar names used by ethnic groups in the region and lack of fixed home addresses or other identifiers that could be used to link primary to relapse cases, while reliability of patient recall of previous treatment is unknown. This means that some VL relapse cases may have been misclassified as primary VL or *vice versa*. Differentiation between relapse and reinfection can only be established by isoenzyme characterisation, which is not possible in programme settings. However, very limited research (in VL patients with HIV co-infection) indicates that cases presenting with a history of previous VL episodes are much more likely to be relapse than re-infection [29]. Another limitation of our analysis is missing data, particularly in years when patient-level data collection was interrupted (2005–2008 and 2011). However, aggregate programme monitoring data was collected during these periods and we think it is sufficiently reliable for monitoring long-term trends. We also noted that VL relapse as a proportion of all cases was similar in programme and patient-level data and both data sources yielded similar trend estimates. The annual percentage change in the best fitting model was supported by marginal statistical evidence and the confidence interval

was quite wide, reflecting substantial year-on-year variation in VL relapse rates. However, the underlying data showing VL relapse increasing from 6% of all VL patients in 2001–2003 to 11% in 2013–2015 and 15% in 2016–2018 gives us a fair degree of confidence that there is a real effect, and the clinical significance of this increase in VL relapse is high.

### Suggestions for the future

Given the limitations of the current retrospective study, MSF is planning a prospective observational study to assess the incidence of relapse (and PKDL) after treatment for primary VL in South Sudan in which patients will be followed up for 12 months. The results of this prospective study will indicate whether drug sensitivity monitoring of anti-leishmanial drugs being used East Africa needs to be considered and will give an indication of the optimal duration of follow-up after treatment to detect the majority of relapses.

The high proportion of relapses presenting to our clinic indicate the importance of encouraging patients to return when symptons re-occur, and of the necessity of close population-level monitoring by national control programs in the region of all VL treatment regimen. However, whether strategies that ensure 6-month or 12-month follow-up can be translated from stable settings such as India or relatively well-resourced effectiveness studies to the South Sudanese context is doubtful at present [30]. Genotyping of parasite isolates from different VL patient categories (primary/relapse, cured/failed) might also reveal predisposing factors to unfavourable treatment outcomes.

Currently under study in 4 East African countries is a 14-day combination regimen of PM (20 mg/kg) in combination with miltefosine (allometric dosing), an oral drug. This combination represents an increased dose of PM, while MF is a drug not previously used in South Sudan. This regimen may provide a viable alternative to SSG/PM. As miltefosine is teratogenic, precautions will need to be taken when administering it to women of child-bearing age and to prevent administration during pregnancy. This study will also generate new data on the efficacy and 6-month relapse rates of SSG/PM, the control arm.

### Conclusion

We have found that increasing incidence of VL relapse in South Sudan does not appear to be explained by changes in patient characteristics or artefact and therefore the effectiveness of current and proposed VL treatments in this setting require urgent and careful evaluation. Drug resistance emerging after decades of SSG/PM combination therapy is a worrying prospect as no alternative treatments are yet available. This underscores the pressing need for new chemical entities that will enable safe and highly effective short-course oral treatments for VL. Until these are available, the new PM/MF regimen that is currently under study can be considered as an alternative treatment option.

### Supporting information

**S1 Table. Médecins Sans Frontières (MSF) visceral leishmaniasis (VL) programme data from Lankien and patient-level data from all MSF sites in South Sudan, 2001–2018.** (PDF)

### Author Contributions

**Conceptualization:** Margriet den Boer, Koert Ritmeijer.

**Data curation:** Gabriel Naylor-Leyland, Simon M. Collin.

**Formal analysis:** Gabriel Naylor-Leyland, Simon M. Collin.

**Supervision:** Margriet den Boer, Koert Ritmeijer.

**Writing – original draft:** Gabriel Naylor-Leyland, Simon M. Collin, Francis Gatluak, Margriet den Boer, Fabiana Alves, Abdul Wasay Mullahzada, Koert Ritmeijer.

**Writing – review & editing:** Gabriel Naylor-Leyland, Simon M. Collin, Francis Gatluak, Margriet den Boer, Fabiana Alves, Abdul Wasay Mullahzada, Koert Ritmeijer.

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
