## [Decision Letter · Decision Letter 0]

2 Jun 2022

Dear Dr. den Boer,

Thank you very much for submitting your manuscript "The increasing incidence of visceral leishmaniasis relapse in South Sudan: a retrospective analysis of field patient data from 2001-2018" for consideration at PLOS Neglected Tropical Diseases. As with all papers reviewed by the journal, your manuscript was reviewed by members of the editorial board and by several independent reviewers. The reviewers appreciated the attention to an important topic. Based on the reviews, we are likely to accept this manuscript for publication, providing that you modify the manuscript according to the review recommendations. 

Sincerely,

Mitali Chatterjee

Associate Editor

Epco Hasker

Deputy Editor

Reviewer's Responses to Questions

**Key Review Criteria Required for Acceptance?**

**Methods**

-Are the objectives of the study clearly articulated with a clear testable hypothesis stated?

-Is the study design appropriate to address the stated objectives?

-Is the population clearly described and appropriate for the hypothesis being tested?

-Is the sample size sufficient to ensure adequate power to address the hypothesis being tested?

-Were correct statistical analysis used to support conclusions?

-Are there concerns about ethical or regulatory requirements being met?

Reviewer #1: Are the objectives of the study clearly articulated with a clear testable hypothesis stated?

Yes

Is the study design appropriate to address the stated objectives?

Yes

Is the population clearly described and appropriate for the hypothesis being tested?

Yes

-Is the sample size sufficient to ensure adequate power to address the hypothesis being tested?

Yes

-Were correct statistical analysis used to support conclusions?

Yes

-Are there concerns about ethical or regulatory requirements being met?

Yes

Reviewer #2: Study is well designed, with clear objectives, methodology, analysis, results, discussion and conclusion. Need to address minor comments provided in the reviewed article.

**Results**

-Does the analysis presented match the analysis plan?

-Are the results clearly and completely presented?

-Are the figures (Tables, Images) of sufficient quality for clarity?

Reviewer #1: Does the analysis presented match the analysis plan?

Yes

-Are the results clearly and completely presented?

Yes

-Are the figures (Tables, Images) of sufficient quality for clarity?

Yes

Reviewer #2: results clearly and completely presented. Need to address minor comments provided in the reviewed article

**Conclusions**

-Are the conclusions supported by the data presented?

-Are the limitations of analysis clearly described?

-Do the authors discuss how these data can be helpful to advance our understanding of the topic under study?

-Is public health relevance addressed?

Reviewer #1: Are the conclusions supported by the data presented?

Yes

-Are the limitations of analysis clearly described?

Yes

-Do the authors discuss how these data can be helpful to advance our understanding of the topic under study?

Yes

-Is public health relevance addressed?

Yes

Reviewer #2: Conclusion is well articulated based on the results presented in the study and need of the VL programme in South Sudan.

**Editorial and Data Presentation Modifications?**

Reviewer #1: Minor Revision

Accept

This is a good study and should be published.

Reviewer #2: Comments are provided in the reviewed article. Need to address those comments to bring better clarity in the paper.

**Summary and General Comments**

Reviewer #1: I have a few points

1. What is the definition of relapse /Reinfection?

2.What are the side effects of the drugs being used for eg SAG , PM,LAMB ?

3.Is 30 mg/kg recommended for LAMB in VL?

4.What were the factors associated with relapse?

Reviewer #2: No general comments.

PLOS authors have the option to publish the peer review history of their article (what does this mean?). If published, this will include your full peer review and any attached files.

Reviewer #1: No

Reviewer #2: Yes: Dr Dhruv Kumar Pandey

Figure Files:

Data Requirements:

Reproducibility:

References

---

## [Editor Report · Decision Letter 1]

23 Jul 2022

Dear Dr. den Boer,

We are pleased to inform you that your manuscript 'The increasing incidence of visceral leishmaniasis relapse in South Sudan: a retrospective analysis of field patient data from 2001-2018' has been provisionally accepted for publication in PLOS Neglected Tropical Diseases.

Best regards,

Mitali Chatterjee

Academic Editor

Epco Hasker

Section Editor

---

## [Editor Report · Acceptance letter]

16 Aug 2022

Dear Mr. Ritmeijer,

We are delighted to inform you that your manuscript, "The increasing incidence of visceral leishmaniasis relapse in South Sudan: a retrospective analysis of field patient data from 2001-2018," has been formally accepted for publication in PLOS Neglected Tropical Diseases.

Best regards,

Shaden Kamhawi

co-Editor-in-Chief

Paul Brindley

co-Editor-in-Chief
